# SeasonDepth: Cross-Season Monocular Depth Prediction Dataset and Benchmark under Multiple Environments

**Hanjiang Hu**[1,2]   **Baoquan Yang**[1]   **Zhijian Qiao**[1]   **Ding Zhao**[2]   **Hesheng Wang**[1]*
[1]Shanghai Jiao Tong University   [2]Carnegie Mellon University
{hanjianghu,dingzhao}@cmu.edu  {yangbaoquan,qiaozhijian,wanghesheng}@sjtu.edu.cn

## Abstract

Different environments pose a great challenge on the outdoor robust visual perception for long-term autonomous driving and the generalization of learning-based algorithms on different environmental effects is still an open problem. Although monocular depth prediction has been well studied recently, there is few work focusing on the robust learning-based depth prediction across different environments, *e.g.* changing illumination and seasons, owing to the lack of such a multi-environment real-world dataset and benchmark. To this end, the first cross-season monocular depth prediction dataset and benchmark *SeasonDepth* [1] is built based on *CMU Visual Localization* dataset. To benchmark the depth estimation performance under different environments, we investigate representative and recent state-of-the-art open-source supervised, self-supervised and domain adaptation depth prediction methods from *KITTI* benchmark using several newly-formulated metrics. Through extensive experimental evaluation on the proposed dataset, the influence of multiple environments on performance and robustness is analyzed both qualitatively and quantitatively, showing that the long-term monocular depth prediction is far from solved even with fine-tuning. We further give promising avenues that self-supervised training and stereo geometry constraint help to enhance the robustness to changing environments.

## 1   Introduction

Outdoor perception and localization for autonomous driving and mobile robotics has made significant progress due to the boost of deep convolutional neural networks [1, 2, 3, 4] in recent years. However, since the outdoor environmental conditions are changing because of different seasons, weather and day time [5, 6, 7], the pixel-level appearance is drastically affected, which casts a huge challenge for the robust long-term visual perception and localization. Monocular depth prediction plays an critical role in the long-term visual perception and localization [8, 9, 10, 11, 12] and is also significant to the safe applications such as self-driving cars under different environmental conditions. Although some depth prediction datasets [13, 14, 15] include some different environments for diversity, however, it is still not clear what kind of algorithm is more robust to adverse conditions and how they influence depth prediction performance. Besides, the generalization of learning-based depth prediction methods on different weather and illumination effects are still an open problem. Therefore,it is indeed needed to build a new dataset and benchmark under multiple environments to systematically study this problem. To the best of knowledge, we are the first to study the generalization of learning-based depth

---

*Corresponding author
[1]Available on https://seasondepth.github.io/.

Submitted to the 35th Conference on Neural Information Processing Systems (NeurIPS 2021) Track on Datasets and Benchmarks. Do not distribute.

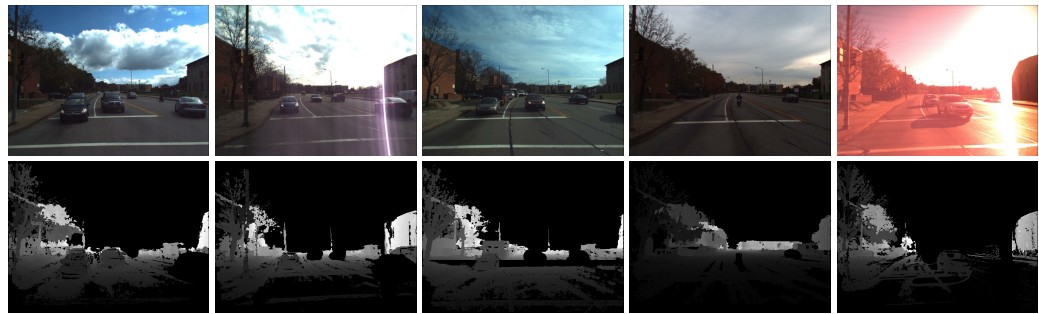

Figure 1: *SeasonDepth* samples with depth map groundtruths under *Cloudy + Foliage*, *Low Sun + Foliage*, *Cloudy + Mixed Foliage*, *Overcast + Mixed Foliage* and *Low Sun + Mixed Foliage*.

prediction under changing environments, which is essential and significant to both robust learning algorithms and practical applications like autonomous driving.

Groundtruth for outdoor high-quality dense depth map is not easy to obtain using LiDAR or laser scanner projection [16, 17, 15], or stereo matching [13, 18, 19], let alone collection under multiple environments. We adopt Structure from Motion (SfM) and Multi-View Stereo (MVS) pipeline with RANSAC followed by careful manual post-processing to build a scaleless dense depth prediction dataset *SeasonDepth* with multi-environment traverses based on the urban part of CMU Visual Localization dataset [6, 20]. Some examples in the dataset are shown in Fig. 1.

For the benchmark on the proposed dataset, several statistical metrics are proposed for the experimental evaluation of the representative and state-of-the-art open-source methods from *KITTI* benchmark [16, 21]. The typical baselines we choose include supervised [1, 22, 23, 24], stereo training based self-supervised [25, 26, 27], monocular video based self-supervised [28, 29, 30, 31, 32] and domain adaptation [33, 34, 35] algorithms. Through thoroughly analyzing benchmark results, we find that no method can present satisfactory performance in terms of $Average$, $Variance$ and $RelativeRange$ metrics simultaneously even if some methods give impressive results on *KITTI* Eigen split [1] and are well fine-tuned on our dataset. We further give the hints of promising avenues to addressing this problem through self-supervised learning or setreo geometry constraint for model trainng. Furthermore, the performance under each environment is investigated both qualitatively and quantitatively for adverse environments.

In summary, our contributions in this work are listed as follows. First, a new monocular depth prediction dataset *SeasonDepth* with same multi-traverse routes under changing environments is introduced through SfM and MVS pipeline and is publicly available. Second, we benchmark representative open-sourced supervised, self-supervised and domain adaptation depth prediction methods from *KITTI* leaderboard on *SeasonDepth* using several statistical metrics. Finally, from the extensive cross-environment evaluation, we point out that which kind of methods are robust to different environments and how changing environments affects the depth prediction to give future research directions. The rest of the paper is structured as follows. Sec. 2 analyzes the related work in depth prediction datasets and algorithms. Sec. 3 presents the process of building *SeasonDepth*. Sec. 4 introduces the metrics and benchmark setup. The experimental evaluation and analysis are shown in Sec. 5. Finally, in Sec. 6 we give the conclusions.

## 2 Related Work

### 2.1 Monocular Depth Prediction Datasets

Depth prediction plays an important role in the perception and localization of autonomous driving and other computer vision applications. Many indoor datasets are built through calibrated RGBD camera [36, 37, 38], expensive laser scanner [17, 39] and web stereo photos [40, 18, 19, 14]. However, outdoor depth map groundtruths are more complex to get, *e.g.* projecting 3D point cloud data onto the image plane [16, 17, 15] for sparse map and using stereo matching to calculate inaccurate and limited-scope depth [13, 14, 18]. Another way to get the depth map is through SfM [41, 24, 42, 15] from monocular sequences. Although this method is time-consuming, it generates pretty accurate relatively-

scaled dense depth maps , which is more general for depth prediction under different scenarios. For

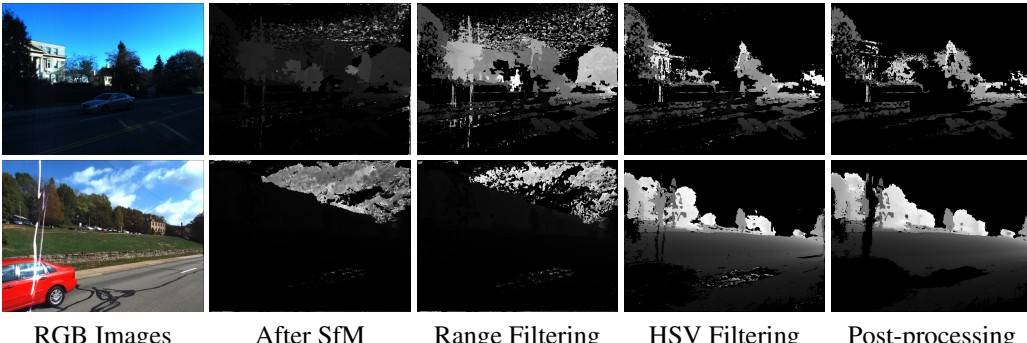

| RGB Images | After SfM | Range Filtering | HSV Filtering | Post-processing |

Figure 2: The illustration of depth map processing.

changing environments, though some real-world datasets [13, 15, 14] include environmental changes, there are still no multi-environment traverses with identical scenarios. Evaluation of robustness across different environments is essential for fairness and reliability. Since graphical rendering is becoming more and more realistic, some virtual synthetic datasets [43, 44, 45, 46] contain multi-environment traverses though the rendered RGB images are still different from real-world ones, where domain adaptation is indispensable and cannot be used to benchmark real-world cross-environment performance. The details of comparison between datasets are shown in Sec. 3.2.

## 2.2 Outdoor Monocular Depth Prediction Algorithms

Monocular depth prediction task aims to predict the dense depth map in an active way given one single RGB image. Early studies including MRF and other graph models [47, 17, 48] largely depend on man-made descriptors, constraining the performance of depth prediction. Afterwards, studies based on CNNs [1, 49, 3] have shown promising results for monocular depth estimation. Eigen *et al.* [1] first predict depth map using CNN model, while [3] introduces fully convolutional neural networks to regress the depth value. After that, supervised methods for monocular depth prediction have been well studied through normal estimation [23, 50], the supervision of depth map and stereo disparity groundtruth [24, 51, 22, 19, 52]. However, since outdoor depth map groundtruths are expensive and time-consuming to obtain, self-supervised depth estimation methods have appeared using stereo geometric left-right consistency [53, 25, 54, 26, 27, 55], egomotion-pose constraint through monocular video [28, 56, 57, 29, 30] and multi-task learning with optical flow, motion and semantics segmentation [58, 59, 31, 32] inside monocular video training pipeline as secondary supervisory signals. Besides, to avoid using expensive real-world depth map groundtruths, other algorithms are trained on synthetic virtual datasets [43, 44, 45, 46] to leverage high-quality depth map groundtruths with zero cost. Such methods [34, 33, 60, 35, 61] confront with the domain adaptation from synthetic to real-world domain only with supervision on virtual datasets for model training.

## 3 SeasonDepth Dataset

Our proposed dataset *SeasonDepth* is derived from CMU Visual Localization dataset [20] through SfM algorithm. The original CMU Visual Localization dataset covers over one year in Pittsburgh, USA, including 12 different environmental conditions. Images were collected from two identical cameras on the left and right of the vehicle along a route of 8.5 kilometers. And this dataset is also derived for long-term visual localization [6] by calculating the 6-DoF camera pose of images with more reasonable categories about weather, vegetation and area. To be consistent with the content of driving scenes in other datasets like *KITTI*, we adopt images from Urban area categorized in [6] to build our dataset. More details about the dataset can be found in Supplementary Material Section 1.

Table 1: Comparison between *SeasonDepth* and Other Datasets

| Name | Scene | Real or Virtual | Depth Value | Sparse or Dense | Multiple Traverses | Different Environments | Dynamic Objects |
|---|---|---|---|---|---|---|---|
| NYUV2 [36] | Indoor | Real | Absolute | Dense | × | × | ✓ |
| DIML [37] | Indoor | Real | Absolute | Dense | × | × | × |
| iBims-1 [38] | Indoor | Real | Absolute | Dense | × | × | × |
| Make3D [17] | Outdoor & Indoor | Real | Absolute | Sparse | × | × | × |
| ReDWeb [18] | Outdoor & Indoor | Real | Relative | Dense | × | × | ✓ |
| WSVD [40] | Outdoor & Indoor | Real | Relative | Dense | × | × | ✓ |
| HR-WSI [19] | Outdoor & Indoor | Real | Absolute | Dense | × | × | ✓ |
| DIODE [39] | Outdoor & Indoor | Real | Absolute | Dense | × | × | × |
| OASIS [42] | Outdoor & Indoor | Real | Relative | Dense | × | × | × |
| 3D Movies [14] | Outdoor & Indoor | Real | Relative | Dense | × | ✓ | ✓ |
| KITTI [16] | Outdoor | Real | Absolute | Sparse | × | × | ✓ |
| CityScapes [13] | Outdoor | Real | Absolute | Dense | × | ✓ | ✓ |
| DIW [41] | Outdoor | Real | Relative | Sparse | × | × | ✓ |
| MegaDepth [24] | Outdoor | Real | Relative | Dense | × | × | ✓ |
| DDAD [29] | Outdoor | Real | Absolute | Dense | × | × | ✓ |
| MPSD [15] | Outdoor | Real | Absolute | Dense | × | ✓ | ✓ |
| V-KITTI [43] | Outdoor | Virtual | Absolute | Dense | ✓ | ✓ | ✓ |
| SYNTHIA [44] | Outdoor | Virtual | Absolute | Dense | × | × | × |
| TartanAir [45] | Outdoor & Indoor | Virtual | Absolute | Dense | ✓ | ✓ | ✓ |
| DeepGTAV [46] | Outdoor | Virtual | Absolute | Dense | ✓ | ✓ | ✓ |
| **SeasonDepth** | **Outdoor** | **Real** | **Relative** | **Dense** | ✓ | ✓ | × |

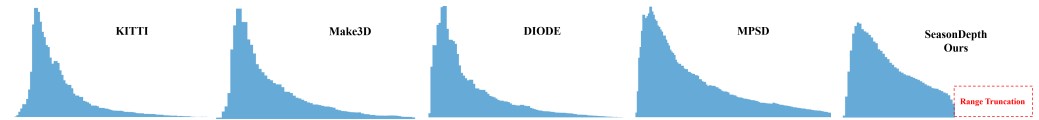

Figure 3: Comparison of relative depth distributions of several datasets.

## 3.1 Depth Dense Reconstruction and Post-processing

We reconstruct the dense model for each traversal under every environmental condition through SfM and MVS pipeline [62], which is commonly used for depth reconstruction [29, 24] and most suitable for multi-environment dense reconstruction for 3D mapping [63, 6] and show advantage on the aspects of high dense quality despite of huge computational efforts compared to active sensing from LiDAR. Specifically, similar to *MegaDepth* [24], COLMAP [64, 62] with SIFT descriptor [65] is used to obtain the depth maps through photometric and geometric consistency from sequential images. Furthermore, we adopt RANSAC algorithm in the SfM to remove the inaccurate values of dynamic objects in the images through effective modification in SIFT matching triangulation based on original COLMAP, where dynamic objects with additional motion besides relative motion to camera do not obey the multi-view geometry constraint and should be removed as noise via RANSAC in bundle adjustment optimization. Since the MVS algorithm generates the depth maps with error pixel values which are out of range or too close, like the cloud in the sky or noisy points on the very near road, we filter those outside the normal range of the depth map.

After the reconstruction, based on the observation of noise distribution in the HSV color space, *e.g.* blue pixels always appear in the sky and dark pixels always appear in the shade of low sun which tend to be noise in most cases, we remove the noisy values in the HSV color space given some specific thresholds. Though outliers are set to be empty in RANSAC, instance segmentation is adopted through MaskRCNN [66] to fully remove the noise of dynamic objects. However, since it is difficult to generate accurate segmentation maps only for dynamic objects under drastically changing environments, we leverage human annotation as the last step to finally check the depth map. Note that since there are often more mis-reconstructed depth pixels around thin objects like branches and poles, we manually filter some of them in the processing for accuracy and reliable evaluation. The data processing is shown in Fig.2 with normalization after each step. More details can be found in Supplementary Material Section 1.1.

## 3.2 Comparison with Other Datasets

The current datasets are introduced in Sec. 2.1. The comparison between *SeasonDepth* and current datasets is shown in Tab. 1. The distinctive feature of the proposed dataset is that *SeasonDepth* contains comprehensive outdoor real-world multi-environment sequences with repeated scenes, just like virtual synthetic datasets [43, 46, 45] but they are rendered from computer graphics and suffer from the huge domain gap. Though real-word datasets [15, 14, 13] include different environments, they lack the same-route traverses under different conditions so they are not able to fairly evaluate the performance across changing environments. Similar to outdoor datasets [41, 24, 42], the depth maps of ours are scaleless with relative depth values, where the metrics should be designed for evaluation as the following section shows. The depth map groundtruths from SfM are dense compared to LiDAR-based sparse depth maps. Besides, since dynamic objects act as noise theoretically for SfM and depth reconstruction, we remove dynamic objects are via RANSAC and instance segmentation but static vehicles are kept with threshold hyperparameters shown in Supplementary Material Tab. 2, which makes the dataset benchmark more reliable and accurate than [29, 24]. And it does not affect the evaluation for driving applications with dynamic objects because it cannot be distinguished whether the objects are dynamic or static given a single monocular image when testing. Consequently, the evaluation on the depth prediction of static objects can reveal the performance of dynamic objects as well although they are not involved in the ground truth.

Besides, the comparison of depth value distribution is shown in Fig. 3. Note that the values of our dataset are scaleless and relative so the x-axes of other dataset are also omitted for fair comparison. We normalize the depth values for all the environments to mitigate the influence of the aggregation from relative depth distributions under different environments to get the final distribution map. The details of implementation can be found in Supplementary Material Section 1.2. From Fig. 3, it can be seen that our dataset also follows the long-tail distribution [67] which is the same as other datasets, with a difference of missing large-depth part due to range truncation during building process in Sec. 3.1.

# 4 Benchmark Setup

The toolkit for the evaluation and benchmark are available here [2].

## 4.1 Evaluation Metrics

The challenge for the design of evaluation metrics lies in two folds. One is to cope with scaleless and partially-valid dense depth map groundtruths, and the other is to fully measure both the depth prediction average performance and the stability or robustness across different environments. Due to scaleless groudtruths of relative depth value, common metrics [21] cannot be used for evaluation directly. Since focal lengths of two cameras are close enough to generate similarly-distributed depth values, unlike [28, 24, 42], we align the distribution of depth prediction to that of depth groundtruths via mean value and variance for fair evaluation. The other key point for multi-environment evaluation lies in the reflection of robustness to changing environments for same-route sequences, which has not been studied in the previous work to the best of our knowledge. We formulate our metrics below.

First, for each pair of predicted and groundtruth depth maps, the valid pixels $D^{i,j}_{valid_{predicted}}$ of the predicted depth map $D_{valid_{predicted}}$ are determined by non-empty valid pixels $D^{i,j}_{valid_{GT}}$ of the depth map groundtruth. And then the valid mean and variance of both $D_{valid_{GT}}$ and $D_{valid_{predicted}}$ are calculated as $Avg_{GT}$, $Avg_{pred}$ and $Var_{GT}$, $Var_{pred}$. Then we adjust the predicted depth map $D_{adj}$ to get the same distribution with $D_{valid_{GT}}$, $D_{adj} = (D_{pred} - Avg_{pred}) \times \sqrt{Var_{GT}/Var_{pred}} + Avg_{GT}$ The examples of adjusted depth prediction are shown in Fig. 4. After this operation, we can eliminate scale difference for depth prediction across datasets, which makes this zero-shot evaluation on *SeasonDepth* reliable and applicable to all the models even though they predict absolute depth values, showing generalization ability on new dataset and robustness across different environments. Denote the adjusted valid depth prediction $D_{adj}$ as $D_P$ in the following formulation. To measure the depth prediction performance, we choose the most distinguishable metrics under multiple environments from commonly-used metrics in [21], *AbsRel* and $\delta < 1.25$ $(a_1)$.

---

[2]Available on https://github.com/SeasonDepth/SeasonDepth.

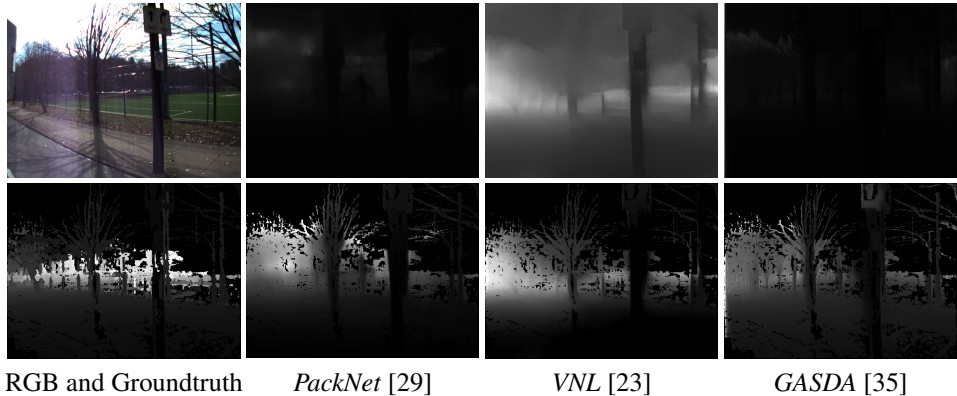

| RGB and Groundtruth | *PackNet* [29] | *VNL* [23] | *GASDA* [35] |

Figure 4: The examples of depth adjustment (from the first to second row) for prediction results.

For environment $k$, we have $AbsRel^k = \frac{1}{n}\sum_{i,j}^{n}\left|D_P{}^k{}_{i,j} - D_{GT}{}^k{}_{i,j}\right|/D_{GT}{}^k{}_{i,j}$ and $a_1^k = \frac{1}{n}\sum_{i,j}^{n}\mathbb{1}(max\{\frac{D_P{}^k{}_{i,j}}{D_{GT}{}^k{}_{i,j}}, \frac{D_{GT}{}^k{}_{i,j}}{D_P{}^k{}_{i,j}}\} < 1.25)$. For the evaluation under different environments, six secondary metrics are derived based on original metrics and statistics, $AbsRel^{avg} = \frac{1}{m}\sum_k AbsRel^k$, $AbsRel^{var} = \frac{1}{m}\sum_k\left|AbsRel^k - \frac{1}{m}\sum_k AbsRel^k\right|^2$, $AbsRel^{relRng} = (max\{AbsRel^k\} - min\{AbsRel^k\})\Big/\frac{1}{m}\sum_k AbsRel^k$ and $a_1^{avg} = \frac{1}{m}\sum_k a_1^k$, $a_1^{var} = \frac{1}{m}\sum_k\left|a_1^k - \frac{1}{m}\sum_k a_1^k\right|^2$, $a_1^{relRng} = (max\{1-a_1^k\} - min\{1-a_1^k\})\Big/\frac{1}{m}\sum_k (1-a_1^k)$, where $avg$ terms $AbsRel^{avg}$, $a_1^{avg}$ and $var$ terms $AbsRel^{var}$, $a_1^{var}$ come from *Mean* and *Variance* in statistics, indicating the average performance and the fluctuation around the mean value across multiple environments.

Considering the depth prediction applications, it should be more rigorous to prevent the fluctuation of better results than that of worse results under changing conditions. Therefore, we use the *Relative Range* terms $AbsRel^{relRng}$, $a_1^{relRng}$ to calculate the relative difference of maximum and minimum for all the environments. *Relative Range* terms for *AbsRel* and $1-a_1$ are more strict than the *Variance* terms $AbsRel^{var}$, $a_1^{var}$ and note that $1-a_1$ instead of $a_1$ is used to calculate $a_1^{relRng}$ to make relative range fluctuation more distinguishable for better methods.

## 4.2 Evaluated Algorithms

Following the category introduced in Sec. 2.2, we have chosen the representative baseline methods together with recent open-source state-of-the-art models on *KITTI* leaderboard [21] to evaluate the performance on the *SeasonDepth* dataset. The evaluated methods include supervised and self-supervised models trained on real-world images, and domain adaptation models trained on virtual synthetic images. More details about the benchmark models including fine-tuning details can be found in Supplementary Material Section 2.1.

For the supervised methods, we choose Eigen *et al.* [1], *BTS* [22], *MegaDepth* [24] and *VNL* [23]. Eigen *et al.* propose the first method using CNNs to predict depth map with scale-invariant loss. *BTS* proposes novel multi-scale local planar guidance layers in decoders for full spatial resolution to get impressive ranked-4th performance. *MegaDepth* introduces an end-to-end hourglass network for depth prediction using semantic and geometric information as supervision. *VNL* proposes the virtual normal estimation which utilizes a stable geometric constraint for long-range relations in a global view to predict depth.

We further choose self-supervised models of stereo training, monocular video training and multi-task learning as secondary signals with video training. Previous work *Monodepth* [25] and two recent work

Table 2: *SeasonDepth* Benchmark Results (↓: Lower Better, ↑: Higher Better, **Best**, Second Best)

| | Method | *KITTI* Eigen Split | | *SeasonDepth*: Average | | Variance$(10^{-2})$ | | Relative Range | |
|---|---|---|---|---|---|---|---|---|---|
| | | *AbsRel* ↓ | $a_1$ ↑ | *AbsRel* ↓ | $a_1$ ↑ | *AbsRel* ↓ | $a_1$ ↓ | *AbsRel* ↓ | $1-a_1$ ↓ |
| Supervised | Eigen *et al.* [1] | 0.203 | 0.702 | 1.093 | 0.340 | 0.346 | **0.0170** | 0.206 | 0.0746 |
| | BTS [22] | **0.060** | **0.955** | 0.677 | 0.209 | 0.539 | 0.0650 | 0.404 | 0.129 |
| | BTS (fine-tuned) | — | — | 0.564 | 0.295 | 0.248 | 0.0943 | 0.309 | 0.151 |
| | MegaDepth [24] | 0.220 | 0.632 | 0.515 | 0.417 | 0.0874 | 0.0285 | 0.200 | 0.107 |
| | VNL [23] | 0.072 | 0.938 | **0.306** | **0.527** | 0.126 | 0.166 | 0.400 | 0.290 |
| Self-supervised Stereo Training | Monodepth [25] | 0.148 | 0.803 | 0.436 | 0.455 | **0.0475** | 0.0213 | 0.198 | 0.104 |
| | adareg [26] | 0.126 | 0.840 | 0.507 | 0.405 | 0.0630 | 0.0474 | **0.178** | **0.0137** |
| | monoResMatch [27] | 0.096 | 0.890 | 0.487 | 0.389 | 0.286 | 0.0871 | 0.414 | 0.160 |
| Self-supervised Monocular Video Training | SfMLearner [28] | 0.181 | 0.733 | 0.693 | 0.265 | 0.151 | 0.0177 | 0.199 | 0.0640 |
| | SfMLearner (fine-tuned) | — | — | 0.485 | 0.455 | 0.412 | 0.103 | 0.405 | 0.241 |
| | PackNet [29] | 0.116 | 0.865 | 0.722 | 0.421 | 0.187 | 0.0705 | 0.186 | 0.155 |
| | Monodepth2 [30] | 0.106 | 0.874 | 0.420 | 0.429 | 0.0848 | 0.0907 | 0.229 | 0.188 |
| | CC [31] | 0.140 | 0.826 | 0.648 | 0.479 | 0.223 | 0.0881 | 0.280 | 0.241 |
| | SGDepth [32] | 0.113 | 0.879 | 0.648 | 0.480 | 0.0987 | 0.0498 | 0.197 | 0.169 |
| Syn-to-real Domain Adaptation | Atapour *et al.* [33] | 0.110 | 0.923 | 0.687 | 0.300 | 0.224 | 0.0220 | 0.231 | 0.0622 |
| | T2Net [34] | 0.169 | 0.769 | 0.827 | 0.391 | 0.399 | 0.0799 | 0.286 | 0.146 |
| | GASDA [35] | 0.143 | 0.836 | 0.438 | 0.411 | 0.121 | 0.0665 | 0.271 | 0.145 |

*adareg* [26], *monoResMatch* [27] are evaluated to present the performance of models trained with stereo geometric constraint. For joint pose regression and depth prediction using video sequences, we test the first method *SfMLearner* [28] and two recent methods *Monodepth2* [30], *PackNet* [29], where *Monodepth2* model also involves stereo geometric information in model training. Besides, we evaluate *CC* [31] with optical flow estimation and motion segmentation, and *SGDepth* [32] with supervised semantic segmentation inside the monocular video based self-supervised framework.

For models trained on the virtual dataset with multiple environments, we evaluate several recent competitive algorithms Atapour *et al.* [33], *T2Net* [34] and *GASDA* [35]. Atapour *et al.* [33] use CycleGAN [68] to train depth predictor with translated synthetic images using virtual groundtruths from DeepGTAV [46]. *T2Net* is a fully supervised method both on *KITTI* and *V-KITTI* dataset and it enables synthetic-to-real translation and depth prediction simultaneously. But *GASDA* is self-supervised for real-world images by incorporating geometry-aware loss through wrapping stereo images together with image translation from synthetic to real-world domain.

# 5 Experimental Evaluation Results

## 5.1 Evaluation Comparison from Overall Metrics

In this section we analyze and discuss what kinds of algorithms are more robust to changing environments by giving several main findings and avenues and their impacts on the performance. The qualitative results of open-source best depth prediction baselines can be found in Tab. 2. To alleviate the impact of dataset bias between *KITTI* and *SeasonDepth*, we adopt one held-out training set to fine-tune one supervised [22] and one self-supervised model [28], which perform poor zero-shot results. Since our dataset does not contain stereo images and share scenarios in V-KITTI dataset, the stereo training based, multi-task training with semantic segmentation and domain adaptation models are omitted to be fine-tuned for fairness.

To make sure the findings and claims are predominantly owing to the different conditions instead of the domain shift, the analysis of fine-tuning is first presented before other critical findings and avenues to this problem. We choose the best results of $Average$ value on SeasonDepth for the fine-tuned models while they still present great limitations on $Variance$ and $RelativeRange$ compared to other baselines or even themselves without fine-tuning. Consequently, fine-tuning helps little to the robustness to changing environments though average performance is improved because of reducing the domain gap, indicating that solely increasing the variability of training data cannot deal with the challenge of environmental changes. After the validation of ineffectiveness of fine-tuned models, to make the evaluation and comparison fair, we draw our conclusion considering all the models regardless they are fine-tuned or not. But one thing for sure is that, all the findings and comparisons below are fair and the performance on $Variance$ and $RelativeRange$ is convincing to purely reflect robustness across different environments since fine-tuning reduces domain gap but does not work for robustness in this case.

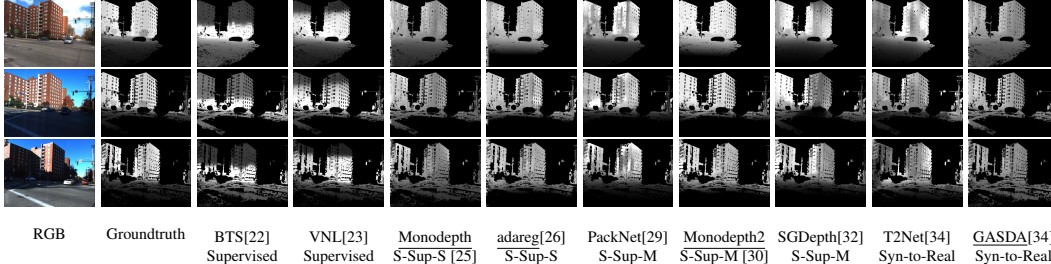

| RGB | Groundtruth | BTS[22] Supervised | VNL[23] Supervised | Monodepth S-Sup-S [25] | adareg[26] S-Sup-S | PackNet[29] S-Sup-M | Monodepth2 S-Sup-M [30] | SGDepth[32] S-Sup-M | T2Net[34] Syn-to-Real | GASDA[34] Syn-to-Real |

Figure 5: Qualitative results for supervised, self-supervised stereo based (S-Sup-S), self-supervised monocular video based (S-Sup-M) and domain adaptation (Syn-to-Real) methods . The conditions from top to down are *S+NF, Apr.* $4^{th}$, *LS+MF, Nov.* $3^{rd}$ and *LS+MF, Nov.* $12^{th}$. Methods denoted with underline are trained with stereo geometry constraint for easier reference and comparison.

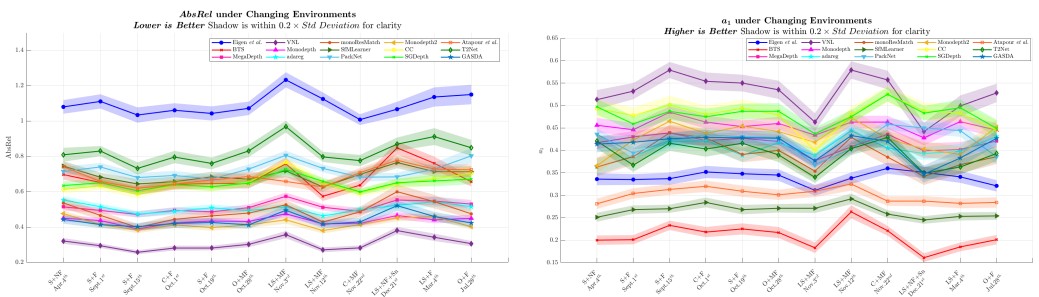

Figure 6: Results on *SeasonDepth* dataset under 12 different environments with dates. The shadows indicate error bars around mean values with $0.2 \times Standard\ Deviation$ for more clarity.

The self-supervised methods show more robustness to different environments compared to supervised methods due to the influence of overfitting from *KITTI* in *SeasonDepth* dataset. Supervised methods suffer from large values of $Variance$ and $RelativeRange$ across multiple environments compared to self-supervised methods, showing that supervised methods are more sensitive to changing environments and even the best fine-tuned model on $Average$ presents poor $Variance$ and $RelativeRange$ performance as well. Besides, although the first proposed several depth prediction methods [1, 25, 28, 33] perform worse than recent methods on *KITTI* and overall $Average$, they show impressive stability to different environments through low $Variance$ and $RelativeRange$.

The second finding is that inside the self-supervised methods, stereo training based methods [25, 26, 27] are more robust to different environments than monocular video training based methods [28, 29] or even with multi-task learning [31, 32] via the comparison on $Variance$ and $RelativeRange$. More broadly, training with stereo geometry constraint clearly helps to improve the robustness to the changing environments compared to those without it for monocular video training based and syn-to-real domain adaptation models, as shown by the quantitative results [30, 35] with light blue shade in Tab. 2 and qualitative results with underline in Fig. 5. Interestingly, the methods with good $Variance$ performance are not consistent with those with good $Average$ performance, which indicates that algorithms tend to work well in specific environments instead of being effective and robust to all conditions, validating the significance of the cross-environment study with *SeasonDepth* dataset and benchmark.

Qualitative results for different types of baselines are shown in Fig. 5. It can be seen that supervised methods *BTS* [22] and *VNL* [23] clearly suffer from overfitting through the predicted pattern where the top and bottom areas are dark while the middle areas are light, even for buildings. Stereo training involved methods with underlines [30, 35] perform continuous depth results for the same entity under all environments, *e.g.* the depth prediction of buildings compared to other self-supervised monocular (S-Sup-M) video based methods [29, 32] and syn-to-real (Syn-to-Real) domain adaptation method [34], validating the improvement of robustness using stereo geometry constraint like quantitative results in Tab. 2. See Supplementary Material Section 2.2 for more qualitative results and analysis.

Table 3: *AbsRel* Results (**Lower Better**) under Each Environment: Mean(Standard Deviation)

| Method | S+NF Apr. 4th | S+F Sept. 1st | S+F Sept. 15th | C+F Oct. 1st | S+F Oct. 19th | O+MF Oct. 28th | LS+MF Nov. 3rd | LS+MF Nov. 12th | C+MF Nov. 22nd | LS+NF+Sn Dec. 21st | LS+F Mar. 4th | O+F Jul. 28th |
|---|---|---|---|---|---|---|---|---|---|---|---|---|
| Eigen et al. [1] | 1.080(0.39) | 1.111(0.40) | 1.034(0.43) | 1.061(0.40) | 1.043(0.40) | 1.072(0.38) | 1.233(0.43) | 1.125(0.37) | 1.008(0.32) | 1.067(0.42) | 1.136(0.54) | 1.150(0.55) |
| BTS [22] | 0.697(0.29) | 0.652(0.24) | 0.605(0.24) | 0.641(0.29) | 0.647(0.27) | 0.646(0.28) | 0.758(0.35) | 0.574(0.27) | 0.637(0.27) | 0.848(0.36) | 0.761(0.38) | 0.657(0.28) |
| MegaDepth [24] | 0.514(0.20) | 0.494(0.16) | 0.471(0.17) | 0.494(0.18) | 0.486(0.18) | 0.510(0.18) | 0.574(0.21) | 0.512(0.18) | 0.489(0.19) | 0.553(0.26) | 0.547(0.25) | 0.530(0.24) |
| VNL [23] | 0.321(0.16) | 0.294(0.13) | 0.257(0.11) | 0.281(0.14) | 0.281(0.13) | 0.302(0.16) | 0.357(0.20) | 0.271(0.14) | 0.282(0.14) | 0.380(0.21) | 0.342(0.21) | 0.306(0.15) |
| Monodepth [25] | 0.450(0.19) | 0.437(0.16) | 0.389(0.14) | 0.424(0.18) | 0.434(0.18) | 0.432(0.16) | 0.475(0.20) | 0.418(0.17) | 0.421(0.16) | 0.465(0.21) | 0.441(0.20) | 0.449(0.20) |
| adareg [26] | 0.553(0.22) | 0.515(0.16) | 0.473(0.18) | 0.489(0.20) | 0.509(0.19) | 0.493(0.19) | 0.515(0.17) | 0.463(0.18) | 0.498(0.20) | 0.523(0.20) | 0.543(0.29) | 0.515(0.25) |
| monoResMatch [27] | 0.536(0.31) | 0.466(0.24) | 0.398(0.19) | 0.444(0.27) | 0.463(0.25) | 0.479(0.31) | 0.526(0.28) | 0.428(0.25) | 0.486(0.28) | 0.600(0.40) | 0.544(0.39) | 0.475(0.26) |
| SfMLearner [28] | 0.745(0.29) | 0.682(0.26) | 0.644(0.27) | 0.657(0.28) | 0.684(0.29) | 0.671(0.28) | 0.718(0.35) | 0.627(0.27) | 0.698(0.27) | 0.765(0.32) | 0.714(0.29) | 0.713(0.31) |
| PackNet [29] | 0.715(0.27) | 0.740(0.23) | 0.680(0.26) | 0.692(0.26) | 0.672(0.24) | 0.728(0.27) | 0.806(0.27) | 0.732(0.22) | 0.682(0.25) | 0.684(0.22) | 0.727(0.36) | 0.803(0.43) |
| Monodepth2 [30] | 0.476(0.18) | 0.414(0.15) | 0.383(0.17) | 0.412(0.17) | 0.396(0.17) | 0.412(0.17) | 0.441(0.23) | 0.380(0.16) | 0.414(0.16) | 0.452(0.20) | 0.459(0.20) | 0.402(0.16) |
| CC [31] | 0.613(0.23) | 0.633(0.23) | 0.587(0.25) | 0.640(0.24) | 0.627(0.27) | 0.652(0.24) | 0.768(0.25) | 0.649(0.23) | 0.593(0.24) | 0.644(0.28) | 0.673(0.34) | 0.703(0.39) |
| SGDepth [32] | 0.635(0.24) | 0.650(0.21) | 0.605(0.23) | 0.640(0.23) | 0.628(0.25) | 0.649(0.24) | 0.726(0.26) | 0.659(0.20) | 0.599(0.19) | 0.651(0.23) | 0.661(0.31) | 0.671(0.29) |
| Atapour et al. [33] | 0.741(0.27) | 0.658(0.22) | 0.619(0.24) | 0.643(0.27) | 0.667(0.27) | 0.686(0.29) | 0.658(0.28) | 0.627(0.29) | 0.708(0.27) | 0.778(0.32) | 0.728(0.29) | 0.724(0.30) |
| T2Net [34] | 0.809(0.39) | 0.830(0.29) | 0.732(0.34) | 0.796(0.35) | 0.760(0.33) | 0.831(0.35) | 0.968(0.33) | 0.797(0.29) | 0.776(0.33) | 0.869(0.37) | 0.912(0.48) | 0.849(0.45) |
| GASDA [35] | 0.443(0.24) | 0.414(0.20) | 0.402(0.21) | 0.420(0.26) | 0.426(0.24) | 0.412(0.22) | 0.495(0.26) | 0.416(0.24) | 0.429(0.24) | 0.521(0.29) | 0.460(0.26) | 0.423(0.26) |

Table 4: $a_1$ Results (**Higher Better**) under Each Environment: Mean(Standard Deviation)

| Method | S+NF Apr. 4th | S+F Sept. 1st | S+F Sept. 15th | C+F Oct. 1st | S+F Oct. 19th | O+MF Oct. 28th | LS+MF Nov. 3rd | LS+MF Nov. 12th | C+MF Nov. 22nd | LS+NF+Sn Dec. 21st | LS+F Mar. 4th | O+F Jul. 28th |
|---|---|---|---|---|---|---|---|---|---|---|---|---|
| Eigen et al. [1] | 0.336(0.14) | 0.335(0.12) | 0.337(0.14) | 0.352(0.14) | 0.348(0.13) | 0.345(0.13) | 0.311(0.12) | 0.338(0.13) | 0.360(0.12) | 0.351(0.13) | 0.341(0.13) | 0.321(0.13) |
| BTS [22] | 0.200(0.11) | 0.201(0.10) | 0.233(0.10) | 0.218(0.11) | 0.225(0.12) | 0.217(0.12) | 0.183(0.12) | 0.263(0.15) | 0.221(0.11) | 0.161(0.10) | 0.185(0.10) | 0.201(0.11) |
| MegaDepth [24] | 0.417(0.14) | 0.430(0.13) | 0.439(0.15) | 0.422(0.16) | 0.427(0.13) | 0.420(0.15) | 0.377(0.13) | 0.408(0.15) | 0.436(0.15) | 0.399(0.17) | 0.402(0.17) | 0.421(0.15) |
| VNL [23] | 0.513(0.21) | 0.532(0.18) | 0.579(0.18) | 0.554(0.20) | 0.550(0.19) | 0.535(0.20) | 0.463(0.20) | 0.579(0.19) | 0.557(0.21) | 0.442(0.19) | 0.499(0.23) | 0.528(0.21) |
| Monodepth [25] | 0.456(0.17) | 0.446(0.15) | 0.485(0.13) | 0.463(0.15) | 0.453(0.14) | 0.460(0.15) | 0.434(0.14) | 0.463(0.14) | 0.463(0.14) | 0.428(0.17) | 0.464(0.16) | 0.445(0.15) |
| adareg [26] | 0.363(0.18) | 0.387(0.14) | 0.419(0.15) | 0.422(0.17) | 0.389(0.14) | 0.417(0.15) | 0.389(0.15) | 0.444(0.16) | 0.405(0.17) | 0.393(0.15) | 0.398(0.16) | 0.431(0.18) |
| monoResMatch [27] | 0.363(0.21) | 0.386(0.18) | 0.439(0.18) | 0.428(0.20) | 0.391(0.17) | 0.400(0.19) | 0.354(0.18) | 0.429(0.20) | 0.385(0.19) | 0.342(0.19) | 0.368(0.20) | 0.386(0.17) |
| SfMLearner [28] | 0.251(0.10) | 0.268(0.09) | 0.270(0.09) | 0.284(0.11) | 0.268(0.11) | 0.271(0.10) | 0.271(0.11) | 0.292(0.12) | 0.258(0.09) | 0.245(0.09) | 0.253(0.09) | 0.254(0.09) |
| PackNet [29] | 0.436(0.13) | 0.394(0.13) | 0.422(0.15) | 0.435(0.15) | 0.430(0.14) | 0.429(0.14) | 0.368(0.13) | 0.403(0.12) | 0.458(0.13) | 0.450(0.13) | 0.444(0.14) | 0.386(0.17) |
| Monodepth2 [30] | 0.366(0.17) | 0.423(0.16) | 0.465(0.19) | 0.438(0.17) | 0.454(0.18) | 0.442(0.16) | 0.418(0.19) | 0.473(0.18) | 0.426(0.17) | 0.403(0.17) | 0.391(0.18) | 0.452(0.16) |
| CC [31] | 0.493(0.19) | 0.478(0.18) | 0.501(0.21) | 0.480(0.20) | 0.494(0.19) | 0.479(0.19) | 0.400(0.15) | 0.480(0.18) | 0.525(0.18) | 0.488(0.19) | 0.483(0.20) | 0.445(0.21) |
| SGDepth [32] | 0.497(0.17) | 0.459(0.16) | 0.487(0.19) | 0.475(0.18) | 0.487(0.17) | 0.487(0.18) | 0.437(0.14) | 0.475(0.15) | 0.525(0.15) | 0.483(0.16) | 0.495(0.18) | 0.449(0.19) |
| Atapour et al. [33] | 0.281(0.12) | 0.304(0.12) | 0.313(0.12) | 0.320(0.13) | 0.309(0.13) | 0.301(0.11) | 0.309(0.13) | 0.325(0.15) | 0.287(0.11) | 0.287(0.11) | 0.282(0.11) | 0.284(0.12) |
| T2Net [34] | 0.421(0.17) | 0.367(0.15) | 0.416(0.17) | 0.403(0.17) | 0.416(0.16) | 0.390(0.16) | 0.340(0.13) | 0.404(0.15) | 0.429(0.17) | 0.349(0.14) | 0.363(0.16) | 0.393(0.17) |
| GASDA [35] | 0.414(0.18) | 0.418(0.16) | 0.426(0.14) | 0.429(0.17) | 0.428(0.16) | 0.427(0.15) | 0.377(0.16) | 0.433(0.18) | 0.420(0.17) | 0.347(0.19) | 0.383(0.19) | 0.427(0.16) |

## 5.2 Performance under Different Environmental Conditions

In this section, we further study how different environments influence the depth prediction results. Different from how different methods perform under multiple environments, this section investigate which environment is the difficult to the current depth prediction models, where Standard Deviation can clearly show that. The detailed results with mean values and standard deviations are shown in Tab. 3 and Tab. 4 and the line chart with shadow error bar in Fig. 6 shows performance in changing environments intuitively. The abbreviations of environments are *S* for *Sunny*, *C* for *Cloudy*, *O* for *Overcast*, *LS* for *Low Sun*, *Sn* for *Snow*, *F* for *Foliage*, *NF* for *No Foliage*, and *MF* for *Mixed Foliage*. From Fig. 6, we can see that although different methods perform differently on *AbsRel* and $a_1$, the influence of some environments is similar for all the methods. Most methods perform well under *S+F, Sept.* $15^{th}$ and *LS+MF, Nov.* $12^{th}$ while dusk scenes in *LS+MF, Nov.* $3^{rd}$ and snowy scenes in *LS+NF+Sn, Dec.* $21^{st}$ pose great challenge for most algorithms, which points out directions for future research and safe applications.

Under these adverse environmental conditions, the promising algorithms can also be found. For the dusk or snowy scenes, domain adaptation methods [33, 34] present impressive robustness due to the various appearances of synthetic images. Besides, for the snowy scenes, self-supervised stereo-based [26, 25, 30] and monocular video training models [31, 32, 29] are less influenced compared to supervised methods. From the error bar and standard deviation in Tab. 3 and Tab. 4, it can be seen that models with larger mean values tend to have larger deviation for each environment, while more adverse environments always result in larger deviations for all algorithms, indicating that adverse environments influence the results of all the methods.

Furthermore, qualitative experimental results are shown in Fig. 7 to show how extreme illumination or vegetation changes affect the depth prediction. We visualize the adjusted results of three overall good methods with robustness to changing environments according to Sec. 5.1 and Tab. 2. From the top two rows, it can be seen that illumination change of low sun makes the depth prediction of tree trunks less clear under the same vegetation condition as green and red blocks show. Also, no foliage tends to make telephone pole and tree trunk less distinguishable by comparing red and green blocks from the last two rows, while the depth prediction of heavy vegetation is difficult as red blocks show on the fourth row given the same illumination and weather condition. More qualitative results can be found in Supplementary Material Section 2.2.

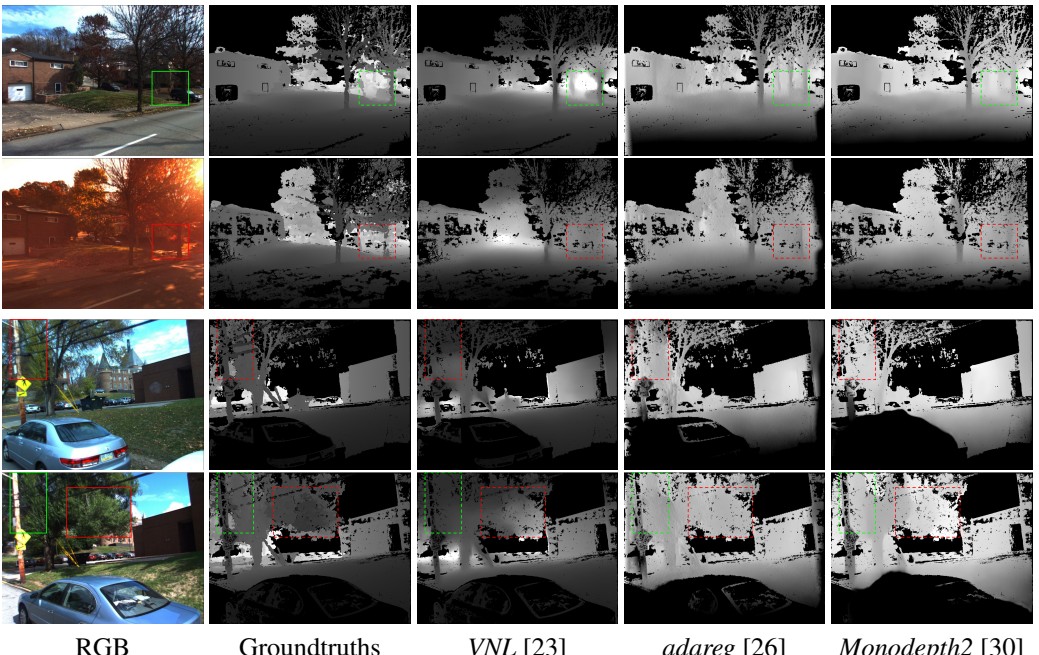

| RGB | Groundtruths | *VNL* [23] | *adareg* [26] | *Monodepth2* [30] |

Figure 7: Qualitative comparison results with illumination or vegetation changes. The conditions from top to down are *C+MF, Nov.* $22^{nd}$, *LS+MF, Nov.* $3^{rd}$, *C+MF, Nov.* $22^{nd}$ and *C+F, Oct.* $1^{st}$. Green blocks indicate good performance while red blocks are for bad results.

## 5.3 Limitation and Discussion

In this section, we discuss the limitation in our work. As mentioned before, our *SeasonDepth* dataset is built based on CMU Visual Localization dataset, which was originally collected for visual localization and contained multiple scenes but without challenging night scenes. Although it is different from the dataset for autonomous driving like *KITTI*, which causes the concern about the evaluation due to the domain gap. But it is acceptable based on the experimental evidence that fine-tuned models will not perform better in terms of $Variance$ and $RelativeRange$. Since dynamic objects are not included in the dataset to ensure accuracy and reliability and it brings about concerns on the driving application. But dynamic object will not hurt to the evaluation of multi-environment depth prediction performance and robustness as shown in Sec. 3.2. For the benchmark, although we try our best to survey and test the open-source representative models as many as possible, it is not possible to involve all the monocular depth prediction methods in our benchmark. So we will release the test set and benchmark toolkit to make up for it. Besides, though some large standard deviations in Tab. 3 and Tab. 4 weaken the credibility and reliability for the performance of methods, the quality of depth map groundtruths is assured so we attribute it to the poor generalization ability of those algorithms since not all the methods present such poor results with too large variances, which cannot be correctly analyzed.

## 6 Conclusion

In this paper, a new dataset *SeasonDepth* is built for monocular depth prediction under different environments. Best open-source supervised, self-supervised and domain adaptation depth prediction algorithms from *KITTI* benchmark are evaluated. From the experimental results, we find that there is still a long way to go to achieve robustness for long-term depth prediction and several promising aspects are given. Self-supervised methods present better robustness than supervised methods to changing environments and stereo geometry involved model training is shown to help to stabilize the cross-environment performance. Through giving hints of how adverse environments influence environments, our findings via the dataset and benchmark will impact the research on long-term robust perception and related application.

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
