# OpenReview forum: "SeasonDepth: Cross-Season Monocular Depth Prediction Dataset and Benchmark under Multiple Environments"
_NeurIPS.cc/2021/Track/Datasets_and_Benchmarks/Round1 — Submitted to NeurIPS 2021 Datasets and Benchmarks Track (Round 1)_

### Official Review · Reviewer_xvJF · 2021-07-04
**Review for Benchmark under Multiple Environments**

**Rating:** 6
**Confidence:** 3

**Strengths:**

Although not as large as KITTI, this dataset (10G) contains a variety of environments for depth estimation for outdoor driving scene.  The process of getting groundtruth dense depth map is well-documented and the process of evaluation for benchmarking is also well written.

Significance of the contribution:
I rank the first contribution of making public a new depth estimation dataset being the most significant one. The 2nd (benchmarking SOTA algorithms) 3rd (analysis of benchmarking results) is much less significant due to concerns to be raised in "Weaknesses" sections. With the comparison to other existing depth estimation dataset shown in table 1, I believe this would fill the gap of real-world, outdoor, dense, multi-traverse, different object gap. The relative depth is a small drawback here though.

Relevance to the broader research community:
This is relevant to the broader research community and is of great interest in computer vision communities like small item detection, range variant detection and auto-dirving. With the help of accurate depth information, lots of down stream tasks can be conditioned on different depth and have better performances.

Accessibility:
I downloaded and inspected the dataset. This dataset is easily accessible and well-documented.


**Weaknesses:**

To me, my biggest concern of the paper lies in the "zero-shot benchmarking" process.

The benchmark and the analysis following the major contribution of releasing a new depth estimation datset is formatted around a center question: Are current SOTA methods accurate and robust enough to different environment changes like season change? The author tried to answer this question by taking the pre-trained models trained on the KITTI dataset (which does not look quite similar to SeasonDepth) and directly applying it to the current unseen dataset.

It would be fine to me if the authors are only comparing the model's robustness by simply comparing their performance (only 2 metrics, a1 and AbsRel) change when dealing with unseen domain (trained on KITTI, test on SeasonDepth), since there is one domain shift here due to "zero-shot".  To make it more specific, my acceptable argument here would be: "Here are the rank of the performance of SOTA methods on KITTI dataset (trained on KITTI, tested on unseen KITTI) on these metrics, here are the rank of the same peroformance metrics of same methods, trained on KITTI but tested on our SeasonDepth dataset. When testing on different environments, their performance downgrade differently compared to KITTI, hence they are different in robustness"

However, what the authors did (at least what I understood) was to use 6 auxiliary metrics to compare and illustrate the "robustness" in the testing domain, which is much less convincing due to there is already a large domain shift from the KITTI domain to the Pittsburg scenary.

To re-iterate, using pre-trained weights from KITTI (domain A) testing on Pittsburg (domain B) is shifting domain large enough that, vairance within domain B (lets say B1, B2 and B3 environments) is hardly captured by using the model trained on domain A data. Instead, to test the model performance robustness on different environmenet (vairance on domain B1, B2, B3), models should be trained on the held-out set of domain B.

Apart from this major concern that I have, I do not have other significant concerns or weaknesses identified.

**Additional Feedback:**

I only have that one major concern, without which I would vote for a good paper. Wish to hear what other reviewers and authors respond.

Updated comments upon reviewing rebuttal and other reviewers' comment: (Jul 19th)
Thank the authors for the extra information provided. The extra experiments are important and helpful in exclude the behavior and effect of the domain shift. However, I don't think the dataset itself, which is the major contribution here, is novel and large enough for me to raise a rating level.

**Clarity:**

The paper is well written and clear. Maybe the abbreviations can be defined more clearly.

**Correctness:**

The claims made in the submission is correct ot me. The dataset is constructed in a sound way and the benchmark is evaluated with no other concern then ones mentioned in the "Weaknesses" sections.

**Documentation:**

The documentation of the data collection and organization is clear. The data is accessible and available on figshare. The benchmark have available details to be reproduced as they used. pre-trained weights from published source on KITTI dataset.

**Ethics:**

No ethic concerns found.

**Relation To Prior Work:**

The discussion of relationship to prior work is sufficient in this paper. Especially the part in section 2.1 the related algorithms. Table 1 captures the landscape of the depth estimation dataset. For the benchmarked models, as they are not the most relevant contribution, the discussion

**Summary And Contributions:**

The authors proposed a new dataset called SeasonDepth (which is derived from a published dataset but with extra dense depth ground truth label provided) with multi-traverse environment for depth estimation problem and benchmarked multiple state-of-the-art depth estimation algorithm of different categories (supervised, self-supervised, synthetic domain adaptation). They used pre-trained models with weights trained on the same dataset (KITTI) without any fine-tuning for all the benchmarkings. They also provdied analysis to the benchmarked result and concluded that depth prediction has a long way to go.

The used a1 and absRel metric for base performance metric and followed-up with statistic metrics based upon these 2 like mean and variance to quantify the variability of these models performance across environments.

---

> ### Author Response · Authors · 2021-07-12
> **Response to Paper71 Reviewer xvJF with fine-tuned results and analysis**
>
> Thank you very much for your thorough reviews and valuable comments on our paper. We address the concerns as follows and feel free to check out our updated paper with changes in blue for easy reference,
>
> - Weaknesses
>
>   Thanks for your clear explanation and insightful comment and suggestion. That's right and makes a lot of sense, the 6 metrics for a1 and AbsRel indeed involve domain shift together with environmental influence.  So, the key point here is how to make the metrics convincing and persuasive  to solely reflect the influence of different environments by alleviating  the domain shift.
>
>   To deal with this concern, we have fine-tuned the supervised and self-supervised models on our held-out training set to reduce the domain gap. We choose the fine-tuned models with the best _Average_ value and update them in Table 2, where it can be seen that although _Average_ is improved, their performance on _Variance_ and _Relative Range_ is still poor. So the metric of _Average_ is clearly influenced by domain shift, but _Variance_ and _Relative Range_ can reflect the influence of different environments no matter using biased pretrained models or non-biased fine-tuned models.
>
>   Consequently, all the findings and comparisons in the paper are fair and the performance on _Variance_ and _Relative Range_ is still convincing and persuasive to purely reflect the robustness across different environments rather than domain bias (since fine-tuning reduces domain gap but does not work for robustness in this case). After the validation of ineffectiveness for robustness of fine-tuned models, to make the evaluation and comparison fair, we draw our conclusion considering all the models regardless they are fine-tuned or not and use _Variance_ and _Relative Range_ to show the robustness to different environments for analysis to reveal the methodological vulnerability and give promising findings and avenues to improve it.  We have clarified this concern and updated the paper in Section 5 and all parts related to experiments. The details regarding fine-tuned models are shown in Supplementary Material Section 2.1.
>
>
>
> - Clarity
>
>   Thanks for your affirmation . We have rephrased some abbreviations and fixed some typos to make it more clear.

---

### Official Review · Reviewer_1YbZ · 2021-07-05
**Good motivation for a new dataset but not very promising**

**Rating:** 5
**Confidence:** 3

**Strengths:**

Strength of this paper:

1. This paper identifies the problem correctly and does make a step towards the goal by constructing a depth-prediction dataset with different season / weather conditions.
2. I like the exhaustive details supplied by the authors, which eases the reproduction for potential readers.
3. The extensive results do reveal some interesting facts on supervised, self-supervised, and domain adaptation approaches.

**Weaknesses:**

I do have some significant concerns on this submission:

1. Dataset. The major concern is the complicated workflow of generating the ground truth depth. Given that there are only very limited number of views available, it is quite unreliable to use MVS to reconstruct depth, even if with careful filtering and manual editing. I agree that LIDAR output is sparse, but in this case it may have better accuracy since it's physically based. Another concern is that most (or even all) dynamic objects are removed. If I read the paper correctly, these two cameras are on a moving vehicle, then doesn't it mean all objects are in relative motion? Besides, from the provided images in the paper I can see some pedestrians/cars are removed but some other are kept. Is there a threshold for making this static vs dynamic decision? The authors are claiming the proposed dataset is mainly serving the autonomous driving and moving robots, and those dynamic objects are clearly the main prediction targets which are however ignored in this dataset.
2. Synthetic vs Real. The above opinions on the dataset make me think if a synthetic dataset is a better way to handle this case. Since the graphical rendering is becoming more and more realistic, and given the fact that you can control the lighting (and possibly seasons) very easily, seems to me it is a win-win choice since it also gives you the absolute correct depth from the renderer without needing any complicated ground-truth generation pipeline.
3. Contribution. Besides the dataset, this paper doesn't propose any solution (even a naive one) to the identified cross-season problem. So based on my understanding the problem may be just resolved by fine-tuning the network on the new data? If so, the contribution of this paper may not reach the standard of this conference track.
4. Evaluation and text. Despite the extensive experiments conducted by the authors, the current submission fails to deliver the clear insights of the finding. The majority of the text just simply describes those three big tables but doesn't summarize the identified patterns in a clear way. This makes me get lost many times when I went through the evaluation section. Some observations are not general to all the cases, like supervised vs self-supervised. I hope authors can summarize their findings in a more concise and concluded way. Moreover, currently most statements are based on the numeric metrics, however sometimes the qualitative zoom-in comparisons can help us readers better understand where, what, and why.

I would say this submission has a large space for improvement before it is acceptable for publication.



**Additional Feedback:**

Please refer to my weakness comments for details.

[Post rebuttal comments]
Score has been raised from 4 to 5 due to the revision. Please see my response for the new comments.

**Clarity:**

The introduction and related work are ok. But the description of the method and evaluation are quite ad-hoc and lack of proper organization.

**Correctness:**

Most of the claims on the dataset look good to me. The claims in the experiment section need to be revisited as some of them are not generalizable to other cases.

**Documentation:**

The authors make a good effort on providing many details on the dataset.

**Ethics:**

I don't have any concerns on that.

**Relation To Prior Work:**

I'm not completely sure but I think it differs in some extent.

**Summary And Contributions:**

This paper introduces a new cross-season dataset for outdoor depth estimation. The evaluation reveals that there are some issues when trying to apply existing depth estimation approaches to different environments as described by the authors. The motivation is solid: the generalization of learning-based methods on different weather and illumination effects are still an open problem, so a high-quality dataset is indeed needed to ensure safe applications such as self-driving cars.

---

> ### Author Response · Authors · 2021-07-12
> **Response to Paper71 Reviewer 1YbZ with explanations**
>
> Thank you very much for your thorough reviews and valuable comments on our paper. We address the concerns as follows and feel free to check out our updated paper with changes in blue for easy reference,
>
> - Dataset.
>
>   In previous work of building dataset, SfM and MVS pipeline is widely used for depth reconstruction [24,29] (in updated version)  and most suitable for multi-environment dense reconstruction for 3D mapping [6,63] (in updated version), which are also somewhat built using original CMU Localization Dataset. Such evidence indicates our image sequences provide enough image views for SfM and MVS, just like previous work using it for dense reconstruction and mapping, which is not worse than sparse physically-based active sensing like LiDAR.
>
>   For the second concern here, relative motion is correct but the static object obeys the same formulation with multi-view geometry constraint in structure from motion, which can be solved through bundle adjustment optimization together with camera pose just as SLAM and our pipeline do. However, the dynamic objects do not obey the geometry constraint because their own motions are involved in addition to the relative motion to camera, so they will become noise for the optimization during depth reconstruction and should be removed through RANSAC. That’s why some pedestrians and cars are removed while others are kept, and the threshold like min_inlier_ratio can be found in Table 2 of Supplementary Material. Though dynamic objects are removed, it doesn’t affect the evaluation for dynamic objects in real applications because it cannot be distinguished whether the objects are dynamic or static given a single monocular image when testing. Consequently, the evaluation of the depth prediction of static objects can reveal the performance of dynamic objects as well although they are not involved in the ground truth. To be honest, we regard this as a kind of limitation in the updated version in Section 5.3 to discuss it. We have clarified it and updated the paper in the first paragraph of Section 3.1, the first paragraph of Section 3.2, and Section 5.3.
>
> - Synthetic vs Real.
>
>   Yes, graphical rendering is becoming more and more realistic, so some virtual synthetic datasets [43, 44, 45, 46] (in the updated version) containing multi-environment traverses appear, even though the rendered RGB images are still different from real-world ones, where domain adaptation is indispensable and cannot be used to benchmark real-world cross-environment performance to study the robustness to changing environment. We have clarified it and updated the paper in the first paragraph of Section 2.1.
>
>
> - Contribution.
>
>   We have fine-tuned supervised and self-supervised models on our training set. Based on the updated experimental results in Table 2, although fine-tuned models perform better on  _Average_ metric they still suffer from _Variance_ and _Relative Range_, indicating that solely increasing the variability of training data and fine-tuning cannot solve the challenge of environmental changes. Therefore, we make a fair comparison and analysis to reveal the methodological vulnerability and give promising findings and avenues to improve the robustness across different environments. The details regarding fine-tuned models are shown in Supplementary Material Section 2.1.  We have clarified this concern and updated the paper in Section 5 and all parts related to experiments.
>
> - Evaluation and text.
>
>   We have made the findings more clear and general in a more concise and concluded way. The self-supervised methods show more robustness to different environments compared to supervised methods even after fine-tuning, showing from _Variance_ and _Relative Range_ metrics. Furthermore, stereo geometry constraint helps to improve the robustness as well. According to your suggestion, we have added qualitative results in Figure 5 to make a zoom-in comparison available, from which our findings can be understood clearly. More details are updated and can be found in Section 5.1. Also, abstract and introduction have also been updated to clarify the experimental findings.
>
> - Correctness
>
>   We have made the findings more clear and more general to give follow-up research directions. The self-supervised methods show more robustness to different environments compared to supervised methods even after fine-tuning from _Variance_ and _Relative Range_ metrics. Furthermore, stereo geometry constraint helps to improve the robustness as well. More details are updated and can be found in Section 5.1. Also, the abstract and introduction have also been updated to clarify the experimental findings.
>
> - Clarity:
>
>   The description of the method and evaluation follow the outline of first building dataset, then preparing benchmark, and finally evaluating the performance. We have made it clear through part of the reorganization.

---

> > ### Comment · Reviewer_1YbZ · 2021-07-20
> > **Response**
> >
> > Thanks for the clarification and revision to the experimental section.
> >
> > Authors have addressed most of my concerns, so I raised score from 4 to 5. The reason why I still lean towards 'below threshold' is due to the lack of contributions. But I'm ok if this paper is accepted since other reviewers seem to be more convinced. I won't be disappointed in any way, this is more like a borderline submission right now.

---

### Official Review · Reviewer_KHEm · 2021-07-06
**Ok dataset with uncertain impact**

**Rating:** 5
**Confidence:** 4
**Correctness:** 1) The claim that existing monocular …

**Strengths:**

1) Dataset focuses on the robustness of Monocular depth estimation under different environmental conditions which is not the focus of existing datasets. However, for real-world use robustness is important.
2) Evaluation of a variety of methods ranging from supervised to self-supervised methods and domain adaptation.
3) Dataset is available and seems to be ready for use (documentation and code are there).

**Weaknesses:**

1) The work purely evaluates methods from KITTI (forward-facing camera) without fine-tuning on SeasonDepth (this dataset) which features cameras looking at approximately 45 degrees to the side. Therefore, data is heavily biased with respect to KITTI in terms of camera view. As a result, it is unclear whether the environmental conditions or the different camera views are responsible for the performance gap. It is likely, the camera view plays a major role in the performance because no fine-tuning on this dataset was proposed. Instead, it is claimed that the different conditions are predominantly responsible for the performance drop.
2) No experiments with fine-tuning on the training set of the proposed dataset. If seasonal changes are a challenge, it might be just enough to add more variety to the training data (add more seasonal variability). In this case, it can be argued that the issue lies with the limited variability of training data rather than a methodological vulnerability.
3) No ground truth for dynamic objects. It is very difficult to compute ground truth for dynamic objects in real-world datasets. However, it is still a limitation because (e.g.) KITTI does have such ground truth and dynamic objects are highly relevant for driving.
4) While the ground truth is relatively dense, it is also often wrong for thinner structures such as poles and branches due to the fact that the reconstruction at those regions fails often.

Final Edit:

1) and 2) have been partially addressed with fine-tuning two models. However, the authors claim that "fine-tuning helps little to the robustness to changing environments though average performance is improved because of reducing the domain gap, indicating that solely increasing the variability of training data cannot deal with the challenge of environmental changes" (line 239 - 242). Still, I am not convinced that this is true because clearly the AbsRel Variance of the finetuned method BTS [22] was reduced by half after supervised fine-tuning. Furthermore, it seems that (according to my understanding reading the rebuttal, comments, paper, and supplementary material) fine-tuning was done in a self-supervised fashion. The problem with the self-supervised fine-tuning of [28] is that it assumes a static environment to compute the loss. The increased variance is most likely due to dynamic objects that violate this assumption instead of any environmental factors. It could well be that supervised fine-tuning of this method would again substantially decrease the variance. In that light, I find that part of the paper unclear and potentially confusing.

**Additional Feedback:**

My current rating is dominated by the claims that are unwarranted due to camera angle (bias in dataset wrt KITTI for evaluation) and lack of training on the dataset for at least a subset of the methods. Overall, the dataset documentation is good and ready for use but the innovation from the dataset point of view is not outstanding. For example, there is no ground truth for moving objects and for night sequences both of which are challenging and relevant in practice.

Final Edit (apart from my previous feedback):

I am unsure whether this paper will have a significant impact on the NeurIPS community and find that there is a novelty in the dataset. However, the novelty is limited. For example, it is not larger, or higher quality than previous datasets nor does it include dramatic (one exception is the snow sequence) environmental changes. Indeed, the idea of seasonal changes for depth prediction is new but it might not be good or novel enough to convince researchers to use the dataset.
Finally, the authors took a great effort to improve the results and presentation of the paper such that I raise my rating to 5.

**Clarity:**

The paper is mostly coherent and cohesive. Some sections require improved formatting (sec 4.1 with a lot of inline equations) or rephrasing (line 287 "sucked results").

**Documentation:**

There is sufficient detail and documentation for use of the dataset. The website provides a brief and concise overview while the GitHub page provides an evaluation script that can be used directly.

**Relation To Prior Work:**

The relation to prior work is clear and summarized in table 1.

**Summary And Contributions:**

This work proposes a dataset for monocular depth prediction in different environmental conditions. The dataset is derived from the CMU Visual Localization dataset by computing depth map via SFM. It features two cameras mounted approximately 45 degrees to the left and right of the heading of a car driving in suburban environments. The paper evaluates multiple methods from the KITTI benchmark on the new dataset without fine-tuning and compares these approaches with common metrics with additional emphasis on their variance.
The contributions are:
- New (scaleless) depth ground truth for a subset of the CMU Visual Localization data
- Evaluation and comparison of various methods on this dataset

---

> ### Author Response · Authors · 2021-07-12
> **Response to Paper71 Reviewer KHEm with experiments of fine-tuned models**
>
> Thank you very much for your thorough reviews and valuable comments on our paper. We address the concerns as follows and feel free to check out our updated paper with changes in blue for easy reference,
>
> - Weaknesses 1
>
>   That's right, the domain shift or bias between KITTI and SeasonDepth exists. To alleviate this domain gap and evaluate the robustness to changing environments, we have fine-tuned the supervised and self-supervised models which heavily suffer from domain gap using our extra held-out training set and made further comparison and analysis after that. The updated experimental results can be seen in Table 2, and fine-tuned details are in Supplementary Material Section 2.1.
>
>     We choose the best results of _Average_ value on SeasonDepth for the fine-tuned models while they still present great limitations on _Variance_ and _Relative Range_. From this observation, fine-tuning helps little to the robustness to changing environments though average performance is improved because of reducing the domain gap. Consequently, the performance on _Variance_ and _Relative Range_ is still convincing and persuasive to purely reflect the robustness across different environments rather than domain bias since fine-tuning reduces domain gap but does not work for robustness in this case.   See updated Section 5.
>
>
> - Weaknesses 2
>
>   According to your suggestion, we have fine-tuned supervised and self-supervised models. Based on the updated experimental results in Table 2, although fine-tuned models perform better on  _Average_ metric they still suffer from _Variance_ and _Relative Range_, indicating that solely increasing the variability of training data cannot deal with the challenge of environmental changes. Therefore, we make a fair comparison and analysis to reveal the methodological vulnerability. The details on fine-tuned models are in Supplementary Material Section 2.1.  See updated Section 5.
>
>
> - Weaknesses 3
>
>  On the one hand, Dynamic objects are noise from SfM methods, so removing them will make the evaluation and benchmark for the dataset more reliable and accurate. On the other hand, it doesn’t affect the evaluation for dynamic objects in real applications because it cannot be distinguished whether the objects are dynamic or static given a single monocular image when testing.  To be honest, we regard this as a kind of limitation in the updated version in Section 5.3 to discuss it. See updates in the first paragraph of Section 3.1, the first paragraph of Section 3.2 and Section 5.3.
>
> - Weaknesses 4
>
>   During the process of building the dataset, since there are often more mis-reconstructed depth pixels around thin objects like branches and poles, we manually filter some of them for accuracy and reliable evaluation. So it looks to be wrong but does not heavily affect the evaluation. See updates in the second paragraph of Section 3.1.
>
> - Correctness 1
>
>   Although the performance within a method for different environments doesn’t vary that much in Table 3, the results for all the methods under some specific environments are similar in Figure 6 (of the updated version), which gives evidence that the environments indeed have a clear impact on the performance. Besides, according to the experimental results of fine-tuned models, _Variance_, _Relative Range_ and _Standard Deviation_ are not caused by camera angle bias which cannot be improved through fine-tuning. That is to say, the difference in camera angle only influences the _Average_ performance but cross-environment performance of _Variance_, _Relative Range_ and _Standard Deviation_ shows the robustness of different environments. See updates in Section 5 and all parts related to experiments.
>
> - Correctness 2
>
>   According to your suggestion, we have fine-tuned the pretrained models on the training split of the proposed dataset and evaluated the best performance of _Average_, as shown in the updated Table 2. From the results, the problem of cross-seasonal monocular depth prediction is still there through the evidence that _Variance_ and _Relative Range_ values are still poor even though _Average_ is improved, validating the claim that cross-seasonal monocular depth prediction is an open problem holds.
>
> - Clarity
>
>  Since the page length is limited, equations in Section 4 still remain inline but we will update them in the camera-ready version if accepted. All the language is polished and rephrased.
>
> - Additional Feedback
>
>     From the dataset point of view, to be honest, we regard no ground truth for moving objects and lack of night scenes as limitations in the updated version in Section 5.3 to discuss it. However, we believe accuracy and reliability are the first place to remove moving objects but it will not hurt evaluation. See updates in the first paragraph of Section 3.1, the first paragraph of Section 3.2 and Section 5.3.

---

> > ### Comment · Reviewer_KHEm · 2021-07-15
> > **Additional Comments and Question**
> >
> > Thank you for your response and additional experiments. Two additional comments from my side:
> >
> > weakness 3: but it matters for approaches that would use multiple images as input.
> >
> > weakness 4: yes, but I have looked at the data myself and saw that this problem still exists with the provided ground truth. It's of course hard to argue whether this is a systematic issue since it is the ground truth itself.
> >
> > is there an explanation why fine-tuning on this dataset increases the variance of [28]?
> >
> > Overall I find the presentation still to be unclear about when we should consider the Dataset to be solved. Some of the presented methods have low variance (e.g. [25, 26, 30]).  And VNL [23] has both a low average error and quite a low variance as well (even without fine-tuning).

---

> > > ### Comment · Program_Chairs · 2021-07-15
> > > **Additional comment**
> > >
> > > Thank you for your additional response comments and question. Here are our additional responses which may dismiss some of your concerns.
> > >
> > > - "weakness 3: but it matters for approaches that would use multiple images as input."
> > >
> > >     For approaches that would use multiple images as input, when testing, the image in the test set is fed to pretrained models one by one so the model still predicts the depth map individually. For the model training or fine-tuning, dynamic objects without ground truths will not be calculated so the supervision mainly focuses on other pixels. And here there is a dilemma that the prediction on these static objects may not generalize to dynamic pixels, so that may be the reason why fine-tuning on this dataset increases the variance of [28], which is fine-tuned using images sequences as input.
> > >
> > > - "weakness 4: yes, but I have looked at the data myself and saw that this problem still exists with the provided ground truth. It's of course hard to argue whether this is a systematic issue since it is the ground truth itself."
> > >
> > >     Yes, that may be a systematic issue for all of the depth maps generated from SfM.
> > >
> > > - "is there an explanation why fine-tuning on this dataset increases the variance of [28]?"
> > >
> > >     As we point out above, the main reason lies in that [28] is monocular video training based method, which needs sequential images as input. But our ground truths lack dynamic objects and the depth prediction of static objects may not well generalize to the dynamic objects. Therefore, when testing, since test images are sent to the network individually to be evaluated, the fine-tuned model cannot tell which object is static and dynamic and becomes even more confused for the depth prediction. As a result, different environments contain different dynamic objects on the same route when testing, so the variance may become larger than before compared to the fine-tuned supervised method BTS[22]. We think it may be not proper and fair to fine-tune the monocular video training based models because of the lack of dynamic objects as ground truths, but the performance of fine-tuned supervised method BTS[22] shows that fine-tuning gives limited help to the robustness of changing environments so most of our claims still hold.
> > >
> > > - "Overall I find the presentation still to be unclear about when we should consider the Dataset to be solved. Some of the presented methods have low variance (e.g. [25, 26, 30]). And VNL [23] has both a low average error and quite a low variance as well (even without fine-tuning)."
> > >
> > >     The goal of this paper is to show that the problem of multi-environment depth prediction exists and current state-of-the-art methods cannot solve it very well. So "when we should consider the Dataset to be solved" is quite hard to define since there are always new methods with better performance on the metrics just like other datasets, KITTI for example. [25, 26, 30] with low variance show that stereo geometry constraint is beneficial to the robustness of changing environments as we claim in the paper. VNL [23] shows a quite low average error but the variance of a1 and _Relative Range_ values are not that low compared to other self-supervised methods, which is also validated qualitatively in Fig. 5 and it cannot give us any hint for this problem from that.

---

### Official Review · Reviewer_t34L · 2021-07-06
**I could not get what are the distinctive features of this dataset/benchmark and what is its potential impact**

**Rating:** 6
**Confidence:** 3
**Correctness:** As far as i can tell, the submissions…

**Strengths:**

The dataset seems to have some advantages existing ones (yet it is not clear to me how important the differences are, see below)

The experimental evaluation considers several methods from the state of the art, yet I cannot assess the impact of the findings derived form the experimental evaluation

**Weaknesses:**

* I think the motivation for the need of this dataset and benchmark is weak, basically, lines 28-38 (whose writing needs to be improved considerably) are the only motivation. This is a critical aspect to evaluate in the paper, how the problem is approached currently? What other uses for the data set the authors foresee (e.g., it could be used for style transfer with GANs), why is the approached problem relevant nowadays? Etc.

* the methods adopted in the pipeline are not described in the paper, while these may be trivial concepts for people working in this subfield, other readers may struggle with these. Also, it is not clear why is this the best method to use for this process, are not other options for this? Why is this pipeline better? What are its benefits and limitations?

* To me is a bit subjective that the dataset is called real, because the new information with respect to the dataset introduced in [2] has been generated by an algorithm,

* Removing dynamic objects is described as an advantage in lines 131-133, I see this more like a limitation, can the authors clarify this?,


* To me is not clear what are the distinctive features of the proposed data set with respect to others shown in Table 1, what makes unique this data set?  Datasets from references  [43, 45, 46] seem similar, or even better than the proposed one.

* From Section 4.1 is not clear to me why no the metrics that have been used for other datasets cannot be used in the proposed dataset/benchmark?

* I think that the experimental evaluation is a valuable contribution from this paper, however, I am not sure what is the impact of the findings? For example, from 5.1, the main finding seems to be that some algorithms perform better than others in different datasets. Overall, I find this kind of expected/obvious, can the authors argue what is the impact of the findings from that section?, do they comprise new results that have not been published elsewhere? In what sense?



**Additional Feedback:**

****** After author' feedback *********

I appreciate the effort authors dedicated to clarify some of my concerns, based on this I have raised my score.

**Clarity:**

The writing is good in general, yet there are many typos and it can be improved. However, I do not think the paper is clear in motivating the need for a dataset considering this particular scenario (approached problem), what are the benefits/limitations with respect to existing datasets and what is the potential impact of the findings derived from the experimental evaluation.

**Documentation:**

Although is not part of documentation, the details on the pipeline used for generating the dataset should be presented in more detail.

**Ethics:**

No ethical concerns were detected

**Relation To Prior Work:**

Somehow, please see my comment on clarity

**Summary And Contributions:**

A dataset for the estimation of depth from monocular images under different environments (seasons) is introduced. The dataset is build on top a previous dataset, applying a pipeline that includes structure from motion. A comprehensive experimental evaluation is performed using existing methods that have been evaluated in other datasets.

---

> ### Author Response · Authors · 2021-07-12
> **Response to Every Review from Paper71 Reviewer t34L**
>
> Thank you very much for your thorough reviews and valuable comments on our paper. We address the concerns as follows and feel free to check out our updated paper with changes in blue for easy reference,
>
> - Weaknesses: "I think ...Etc."
>
>  The motivation lies in both applications and research. First, monocular depth prediction plays a critical role in the long-term visual perception and localization and is also significant to the safe applications such as self-driving cars under different environmental conditions. Second, the generalization of learning-based depth prediction methods on different weather and illumination effects is still an open problem. Therefore, it is indeed needed to build a new dataset and benchmark under multiple environments to systematically study this problem.  The motivation is reorganized and updated in Section 1 and abstract.
>
> - Weaknesses: "The methods ...limitations? "
>
>  Because of the limitation of page length, we leave the detailed description of the method pipeline in the Supplementary Material Section 1.1, which could help readers to familiarize the subfields. Our method in the pipeline, SfM and MVS pipeline, is commonly used for depth reconstruction [24,29] (updated)  and most suitable for multi-environment dense reconstruction for 3D mapping [6,63] (updated).  COLMAP with SIFT descriptor shows clear advantages on the aspects of high dense quality despite of the limitations of huge computational efforts compared to active sensing based methods like LiDAR. More updated details on the comparison and motivation of different dataset can be found in Section 3.1 and 3.2.
>
>
> - Weaknesses: "To me is a bit subjective...  an algorithm"
>
>   Yes, our depth ground truth is generated by an algorithm, but what we call real is that the RGB images come from our daily life, and they are real-world, instead of rendered images from computer graphics which suffer from huge domain gap to the real world. We have clarified and updated our work in the first paragraph of Section 3.2.
>
> - Weaknesses: "Removing dynamic ...this?"
>
>  On the one hand, Dynamic objects are noise from SfM methods, so removing them will make the evaluation and benchmark for the dataset more reliable and accurate. On the other hand, it doesn’t affect the evaluation for dynamic objects in real applications because it cannot be distinguished whether the objects are dynamic or static given a single monocular image when testing. Consequently, the evaluation of the depth prediction of static objects can reveal the performance of dynamic objects as well although they are not involved in the ground truth. To be honest, we regard this as a kind of limitation in the updated version in Section 5.3 to discuss it. We have clarified it and updated the paper in the first paragraph of Section 3.1, the first paragraph of Section 3.2, Section 5.3.
>
> - Weaknesses: "To me... proposed one. "
>
>      The uniqueness of our dataset is that ours contains real-world RGB images with multiple environments under the same routes, while [43, 45, 46] are only synthetic images rendered from computer graphics which has a huge domain gap to the real world. We have clarified and updated our work in the first paragraph of Section 3.2.
>
> - Weaknesses: "From Section 4.1 ...benchmark? "
>
>     The first point here is that our  ground truth value is scaleless relative and not scaled absolute like KITTI so we cannot directly use the metrics for dataset with absolute values. The second is that to the best of our knowledge, our work is the first to evaluate depth prediction performance under the same routes but different environments, so statistics like variance or range is naturally used as new metrics to reflect the robustness to changing environments for same-route sequences, while metrics for other dataset are not appropriate in this case. We have clarified and updated our work in the first paragraph of Section 4.1.
>
> - Weaknesses: "I think ...sense?"
>
>      We have made the findings more clear and general to give follow-up research directions. The self-supervised methods show more robustness to different environments compared to supervised methods even after fine-tuning from _Variance_ and _Relative Range_ metrics. Furthermore, stereo geometry constraint helps to improve the robustness as well. The impact of the findings lies in the boost of future work on the topic of robust perception through these promising ways. We do not have other submissions regarding this topic right now. More details are updated and can be found in Section 5.1. Also, abstract and introduction have also been updated to clarify the experimental findings.
>
> - Clarity
>
>      We have fixed a few typos in the updated version. The motivation and the impact of the findings are reorganized and clarified. Please see the updated abstract, Section 1,Section 2.1, Section 3.2 and Section 5 for more details.
>
>
> - Documentation
>
>      The details on the pipeline can be found in Supplementary Material Section 1.1.

---

### Official Review · Reviewer_X8Db · 2021-07-06
**A cross-season monocular depth prediction datastet derived from the CMU Visual Localization dataset, comprehensive evaluation on several SOTA methods, and proposal of new metrics**

**Rating:** 7
**Confidence:** 3
**Correctness:** Seems ok.
**Clarity:** The paper will benefit from an edit b…

**Strengths:**

The dataset is a valuable contribution and along with the toolkit which the authors have provided, should spark followup research in this area.

The depth of evaluation is also appreciated with comparisons of different (pre-trained) methods.


**Weaknesses:**

The dataset is a derived from the CMU Visual Localization dataset using SfM which reduces the novelty of the work.

While the breadth of evaluation (number of model variations) is commendable, it is hard to obtain any actionable insights from the experiments regarding promising avenues of future exploration. Some commentary of how follow-up research directions may have been useful.

The authors also largely rely on pretrained models for their experiments.

**Additional Feedback:**

** Update 7/19 **

The authors have done a commendable job in responding to reviewer comments.
While my concerns on the somewhat incremental nature of the work remain (which cannot be addressed during a rebuttal phase), I have raised my score as the authors have addressed the other concerns I raised.



**Documentation:**

There is some limited documentation on github regarding the organization of the dataset and how to use the toolkit. More detailed documentation, instructions, and steps to reproduce the paper results would be beneficial.

**Ethics:**

I don't have any concerns about ethics.

**Relation To Prior Work:**

Seems ok.

**Summary And Contributions:**

This paper introduces a new cross-season monocular depth-prediction dataset which contains a large(r) variety of environments under different seasonal conditions. The dataset is a derivation of the CMU Visual Localization dataset, obtained using SfM.

The authors evaluate the performance of their dataset on a large set of pre-trained SOTA models, which represent the different model categories including supervised methods, self-supervised, and domain adaptation methods. New metrics are also proposed to provide further insight into the performance of the models on the new dataset.

The experimental results showcase the challenge of the different seasons and adverse environmental conditions on the performance of SOTA models, with some preliminary insights into what model categories may be potential candidates for refinment.

---

> ### Author Response · Authors · 2021-07-12
> **Response to Every Review from Paper71 Reviewer X8Db**
>
> Thank you very much for your thorough reviews and valuable comments on our paper. We address the concerns as follows and feel free to check out our updated paper with changes in blue for easy reference,
>
> - The dataset is a derived from the CMU Visual Localization dataset using SfM which reduces the novelty of the work.
>
>     **Our reply:** Thanks for your comment. In previous work of building dataset, deriving from other dataset is commonly used and acceptable for efficiency and high quality. Specifically, SfM and MVS pipeline is widely used for depth reconstruction [24,29] (in updated version)  and most suitable for multi-environment dense reconstruction for 3D mapping [6,63] (in updated version). About the novelty of the work, we adopt new and complex processing techniques  in the SfM pipeline, which is different from previous work, check out more updated details in Section 3.1 and Supplementary Material Section 1.1.
>
> - While the breadth of evaluation (number of model variations) is commendable, it is hard to obtain any actionable insights from the experiments regarding promising avenues of future exploration. Some commentary of how follow-up research directions may  have been useful.
>
>     **Our reply:** Thanks for your comment and suggestion. We have made the findings more clear and general to give follow-up research directions. The self-supervised methods show more robustness to different environments compared to supervised methods even after fine-tuning from _Variance_ and _Relative Range_ metrics. Furthermore, stereo geometry constraint helps to improve the robustness as well. More details are updated and can be found in Section 5.1. Also, abstract and introduction have also been updated to clarify the experimental findings.
>
>
> - The authors also largely rely on pretrained models  for their experiments.
>
>     **Our reply:** Thanks for your comments. We have fine-tuned supervised and self-supervised models on our training set. Based on the updated experimental results in Table 2, although fine-tuned models perform better on  _Average_ metric but they still suffer from _Variance_ and _Relative Range_, indicating that solely increasing the variability of training data cannot deal with the challenge of environmental changes. Therefore, we make fair comparison and analysis based on both pretrained models and fine-tuned models to give promising findings and avenues  to improve the robustness across different environments. The details regarding fine-tuned models are shown in Supplementary Material Section 2.1.  We have clarified this concern and updated the paper in Section 5 and all parts related to experiments.
>
> - The paper will benefit from an edit by a native english speaker
>
>     **Our reply:** Thanks for your suggestion. We have polished the expression and language.
>
>
> - There is some limited documentation on github regarding the organization of the dataset and how to use the toolkit. More detailed documentation, instructions, and steps to reproduce the paper results would be beneficial.
>
>     **Our reply:** Thanks for your comment. The organization of the dataset and how to use the toolkit is shown on [https://github.com/SeasonDepth/SeasonDepth](https://github.com/SeasonDepth/SeasonDepth) and [https://github.com/SeasonDepth/SeasonDepth/tree/master/dataset_info](https://github.com/SeasonDepth/SeasonDepth/tree/master/dataset_info). The steps to reproduce the paper can be found in much more details in the Supplementary Material Section 2.1.

---

### Author Response · Authors · 2021-07-12
**Update for our paper and supplementary material**

Dear reviewers and chairs,

Thanks a lot for your efforts and time. After hard work for the past several days, now we give our comprehensive response of reviews to every reviewer. The biggest update this time is that we conducted two important fine-tuned experiments to dismiss the main concern of domain bias from almost all the reviewers, together with other responses like dynamic objects. Please check out the response to each reviewer and our updated part in our paper is denoted in blue for easy reference. Hope our responses can dismiss your concern and look forward to your feedback or discussion. Thanks once again for the valuable reviews!

Best,
NeurIPS 2021 Track Datasets and Benchmarks Round1 Paper71 Authors

---

### Note · ~Hanjiang_Hu1 · 2021-06-11

Training set v1 and the fine-tuned models are released. Updated dataset website: https://seasondepth.github.io/ ; Updated benchmark toolkit repo: https://github.com/SeasonDepth/SeasonDepth ; Updated detailed dataset information README: https://github.com/SeasonDepth/SeasonDepth/tree/master/dataset_info

---

### Decision · Program_Chairs · 2021-07-26

**Decision:**

Reject

**Comment:**

The paper received some critiques regarding the quality/novelty of the new dataset as well as regarding its analysis. The authors did a great job addressing all reviewers concerns. Overall, the data provided looks very useful to the community but the paper still needs some improvement to be ready for publication. Unfortunatelly it can not be accepted in its current for for round 1. We encourage authors to carefully improve current version of the paper and submit it to round 2.